# Ring Finger 149-Related Is an FGF/MAPK-Independent Regulator of Pharyngeal Muscle Fate Specification

**DOI:** 10.3390/ijms24108865

**Published:** 2023-05-16

**Authors:** Burcu Vitrinel, Christine Vogel, Lionel Christiaen

**Affiliations:** 1Center for Genomics and Systems Biology, Department of Biology, New York University, New York, NY 10003, USA; 2Center for Developmental Genetics, Department of Biology, New York University, New York, NY 10003, USA; 3Michael Sars Centre, University of Bergen, P.O. Box 7800, 5020 Bergen, Norway

**Keywords:** heart, head muscles, post-transcriptional regulation, signaling, fate specification

## Abstract

During embryonic development, cell-fate specification gives rise to dedicated lineages that underlie tissue formation. In olfactores, which comprise tunicates and vertebrates, the cardiopharyngeal field is formed by multipotent progenitors of both cardiac and branchiomeric muscles. The ascidian *Ciona* is a powerful model to study cardiopharyngeal fate specification with cellular resolution, as only two bilateral pairs of multipotent cardiopharyngeal progenitors give rise to the heart and to the pharyngeal muscles (also known as atrial siphon muscles, ASM). These progenitors are multilineage primed, in as much as they express a combination of early ASM- and heart-specific transcripts that become restricted to their corresponding precursors, following oriented and asymmetric divisions. Here, we identify the primed gene ring finger 149 related (*Rnf149-r*), which later becomes restricted to the heart progenitors, but appears to regulate pharyngeal muscle fate specification in the cardiopharyngeal lineage. CRISPR/Cas9-mediated loss of *Rnf149-r* function impairs atrial siphon muscle morphogenesis, and downregulates *Tbx1/10* and *Ebf*, two key determinants of pharyngeal muscle fate, while upregulating heart-specific gene expression. These phenotypes are reminiscent of the loss of FGF/MAPK signaling in the cardiopharyngeal lineage, and an integrated analysis of lineage-specific bulk RNA-seq profiling of loss-of-function perturbations has identified a significant overlap between candidate FGF/MAPK and *Rnf149-r* target genes. However, functional interaction assays suggest that *Rnf149-r* does not directly modulate the activity of the FGF/MAPK/Ets1/2 pathway. Instead, we propose that *Rnf149-r* acts both in parallel to the FGF/MAPK signaling on shared targets, as well as on FGF/MAPK-independent targets through (a) separate pathway(s).

## 1. Introduction

During vertebrate development, the heart arises from distinct first and second heart fields [1,2,3]. Clonal analyses have shown that first and second heart field progenitor cells arise from independent pools of multipotent *Mesp1*+ progenitors [4,5]. However, common pools of progenitors give rise to the second heart field and the branchiomeric/pharyngeal muscles [6,7], referred to as the cardiopharyngeal lineages. 

Here, we leveraged the simplicity of the cardiogenic lineage in *Ciona*, a simple tunicate among the closest living relatives to vertebrates [8,9], to study cardiopharyngeal cell-fate choices. *Ciona* allows us to study the conserved early stages of cardiopharyngeal development with exceptional spatial and temporal resolution, and therefore has emerged as a suitable model organism to understand developmental fate choices between cardiac and pharyngeal muscle cells [10]. In *Ciona*, early lineage commitment typically restricts the competence of progenitors prior to lineage amplification by proliferation; in contrast, multipotent progenitors are amplified prior to fate specification in mammalian embryos [11]. Similar to their vertebrate counterparts, the cardiopharyngeal lineage of *Ciona* stems from multipotent progenitors, the trunk ventral cells (TVCs), which emerge from *Mesp*+ mesodermal progenitors. The TVCs are induced by fibroblast growth factor/mitogen-activated protein kinase (FGF/MAPK) signaling and migrate as bilateral pairs of cells, until the left and right pairs meet at the ventral midline and resume cell divisions (Figure 1a) [12,13,14,15]. Late TVCs undergo oriented and asymmetric divisions that produce first heart precursors (FHPs) and secondary TVCs (STVCs), which then give rise to second heart precursors (SHPs) and pharyngeal muscle precursors (also known as atrial muscle founder cells, ASMFs) (Figure 1a). STVCs activate *Tbx1*/*10*, the homolog of human *TBX1* that is required for cell-specific expression of the pharyngeal muscle determinant *Ebf* [14,16,17]. The FGF/MAPK signaling pathway is a key regulator of successive cardiopharyngeal fate decisions in *Ciona*. FGF/MAPK signaling is required for the induction of first and second-generation multipotent cardiopharyngeal progenitors (also known as TVCs, and STVCs), and for the subsequent activation of the siphon/pharyngeal muscle program in their corresponding progenitors [18,19].

Over the past decade, we have extensively documented gene expression dynamics, and begun to decipher the underlying gene regulatory networks that govern early cardiopharyngeal development in *Ciona*. A key feature of the transcriptome dynamics that determine cardiopharyngeal transitions is multilineage priming, whereby multipotent cardiopharyngeal progenitors co-express early key regulators of the cardiac- and pharyngeal muscle-specific programs [17,22]. We surmise that multilineage transcriptional priming, while contributing to multipotency, also poses a challenge for subsequent fate specification following cell divisions, as fate-restricted progenitors inherit gene products that belong to the alternative fates, and might interfere with commitment to a cardiac or pharyngeal muscle identity. For instance, single cell RNA-seq datasets have indicated that first and second heart precursors inherit pharyngeal muscle-specific mRNAs that are downregulated with varying dynamics after cell division and upon commitment to a cardiac identity [22]. We thus hypothesize that cell-type-specific post-transcriptional regulatory mechanisms contribute to early cardiopharyngeal development, by remodeling inherited transcriptomes and proteomes upon fate specification and commitment. 

Here, we focused on candidate post-transcriptional regulators showing differential gene expression in the cardiopharyngeal lineage and identified *ring finger protein 149 related* (hereafter referred to as *Rnf149-r*), a previously uncharacterized gene, as a necessary determinant of pharyngeal muscle identity. *Rnf149-r* is a transcriptionally primed heart marker in the cardiopharyngeal lineage [22] that also encodes the postplasmic RNA (also known as posterior end mark, PEM) known as *Pen-1* [23]. *Rnf149-r*/*Pen-1* is dynamically expressed in various tissues during embryogenesis, including the central nervous system, the notochord, and the epidermis, further suggesting pleiotropic functions. The predicted structure of the Rnf149-r protein revealed an atypical RNF organization, with a protease-associated domain, but lacking the catalytic RING domain. In the cardiopharyngeal lineage, CRISPR/Cas9-mediated loss of *Rnf149-r* function disrupted pharyngeal muscle specification, most likely through the inhibition of *Ebf* gene expression. The effects of *Rnf149-r^CRISPR^* partially phenocopied the loss of FGF/MAPK signaling, including a significant overlap of dysregulated genes from bulk RNA-seq experiments on FACS-purified cells. Finally, functional interaction assays suggested that *Rnf149-r* acts in parallel to the FGFR/MEK/Ets1/2 pathway upstream of *Ebf* activation, thus revealing the existence of FGF/MAPK-independent regulatory inputs into pharyngeal muscle specification.

## 2. Results

### 2.1. CRISPR/Cas9-Mediated Mutagenesis Identifies the Pharyngeal Muscle Determinant Rnf149-r 

We have previously observed extensive multilineage transcriptional priming in multipotent cardiopharyngeal progenitors [22]. This led us to hypothesize that lineage-specific post-transcriptional regulatory mechanisms contribute to remodeling transcriptomes and proteomes during heart vs. pharyngeal muscle fate decisions. In order to identify candidate post-transcriptional regulators, we cataloged genes encoding proteins annotated as RNA-binding and/or involved in ubiquitination pathways (GO terms GO:0003723 and GO:0016567, respectively), using the ANISEED database of GOSlim annotations, curated using data from InterPro and UniProt (Appendix A) [24,25,26]. We integrated this table, containing 945 candidate genes, with previous cardiopharyngeal lineage-specific single-cell RNA-seq (scRNA-seq) data to identify candidate heart- and pharyngeal-muscle-specific post-transcriptional regulators [22] (Appendix A). In a pilot approach, we selected 14 cardiopharyngeal genes that encode either ubiquitin ligase-related proteins (*Rnf149-r*, *Asb2*, *Bag3/4*, *Rbms1/2/3*) or RNA-binding proteins (*Nova*, *Rbfox1/2/3*, *Rbms1/2/3*, *Rbm24/38*, *Qki*, *Ube2ql1*, *Pcbp3*, *Ube2j1*, *Otud3*, *Psmd14*). We performed dual fluorescent in situ hybridization and immunohistochemistry (FISH-IHC) to verify their predicted expression in the heart and/or pharyngeal muscle precursors (Appendix A). We conducted lineage-specific loss-of-function analyses using the CRISPR/Cas9 system to target 6 of 15 candidate regulators [27,28,29,30]. We collected swimming larvae (stage 29/30; 26 h post-fertilization at ~18 °C), and scored the morphology of the cardiopharyngeal lineage, notably the presence or absence of the conspicuous pharyngeal/atrial siphon muscle (ASM) rings (Figure 1b,c; [14]). We targeted the neurogenic bHLH-factor-coding gene *neurogenin* (*Neurog^CRISPR^*) as a control, since it is neither expressed in the cardiopharyngeal mesoderm nor thought to be involved in cardiopharyngeal development [30]. We used CRISPR reagents targeting the pigment cell-specific marker tyrosinase (*Tyr^CRISPR^*) as an alternative control condition throughout the manuscript.

Lineage-specific CRISPR/Cas9-mediated mutagenesis of *ring finger protein 149 related* (*Rnf149-r*) showed the most penetrant phenotype, characterized by disrupted pharyngeal muscle morphogenesis (Chi-square test, *p*-value < 0.0001; Figure 1d). *Rnf149-r* mutagenesis caused pharyngeal muscle morphogenesis defects, whereby the cells failed to migrate toward the atrial siphon placode (ASP) and form the atrial siphon muscle (ASM) rings or crescents observed in control larvae [14]. ASM rings were present in control conditions targeting either *Neurog* or *tyrosinase* (*Tyr*), which is also inactive in the cardiopharyngeal lineage (Figure 1b–e). In order to confirm the specificity of the *Rnf149-r^CRISPR^* phenotype, we targeted *Rnf149-r* using 2 sgRNAs targeting different positions in the coding sequence (Appendix A). These sgRNAs produced similar pharyngeal muscle morphogenesis defects, whether used in combination or separately, indicating that both reagents contribute to the phenotype and are specific to *Rnf149-r* (Appendix A). In order to further ascertain specificity, we expressed the CRISPR/Cas9 reagents alongside a rescue construct, consisting of a CRISPR/Cas9-resistant form of an *Rnf149-r* cDNA, with mutations in the protospacer adjacent motifs (PAMs) of both of the sgRNAs used (*Rnf149-r^mut^*), and expressed it under the control of the cardiopharyngeal progenitor-specific *Foxf* enhancer [31]. Remarkably, the proportion of larvae showing signs of normal ASM morphogenesis increased from ~20% in *Rnf149-r^CRISPR^* animals to ~70% following co-expression of *Rnf149-r*^mut^ (Figure 1e). Circumstantial evidence suggested that *Rnf149-r^mut^* overexpression did not cause any overt phenotype, and we did not pursue these experiments further. Taken together, these data indicate that the observed pharyngeal muscle phenotype is specifically caused by the loss of *Rnf149-r* function in the cardiopharyngeal lineage.

Consistent with a potential role in cardiopharyngeal development, the uncharacterized gene *Rnf149-r* is transcriptionally primed in multipotent progenitors at the tailbud stage (stage 22), and restricted to the heart progenitors in swimming larvae (Appendix A) [22]. In order to understand whether the phenotype is primarily caused by cellular behavior or fate-specification defects, we assayed the expression of the essential ASM determinant and specific marker *Ebf*, in hatching larvae (stage 26; [14,17,32]). The ASM-specific factor *Ebf* is necessary and sufficient to suppress the heart program and impose the pharyngeal muscle fate in the cardiopharyngeal lineage [14,17]. *Rnf149-r* mutagenesis caused a lineage-specific loss of *Ebf* expression, which typically produced ectopic cardiac specification and ASM fate-specification defects, thus abolishing migration toward the ASP (Figure 2a,b) [17,28].

As we observed phenotypic defects at 18 and 26 hpf, we asked whether earlier cardiopharyngeal development is affected by the loss of *Rnf149-r* function. We tested TVC migration and expression of the TVC marker *Hand-r* at the late tailbud stage (12 hpf at 18 °C). However, we did not observe any difference between the experimental and control animals (Appendix A). In light of these results, we propose that *Rnf149-r* function is necessary for the transition to the pharyngeal muscle fate from a multipotent cardiopharyngeal progenitor state.

### 2.2. Rnf149-r Encodes an Atypical Ubiquitin Ligase-Related Protein

We identified *Rnf149-r* for CRISPR/Cas9 mutagenesis as a candidate post-transcriptional regulator because it was annotated as a RING-finger-domain-containing protein, which typically comprise E3-ubiquitin ligases. However, closer inspection indicated that the predicted *Ciona* Rnf149-r protein lacks a RING domain but contains a protease-associated (PA) domain (Appendix A). PA domains in humans and other higher vertebrates can co-exist with RING domains, as well as other functionally active domains, such as EGF, RZF family, and transferrin receptor domains [33]. 

Similar to other organisms, the *Ciona* genome encodes a variety of PA-domain-containing proteins. This domain in *Ciona* occurs as the only defined domain in two predicted proteins, including Rnf149-r. The other PA-domain proteins also contain associated glycosidase domains, Zn-independent exopeptidase domains, transferrin receptor-like dimerization domains, and/or RING domains (Appendix A). Our sequence analyses showed that Rnf149-r has one homolog with a similar domain architecture in *Ciona*, Rnf150, prompting us to hypothesize that Rnf149-r may act as a natural dominant-negative inhibitor of Rnf150 function. However, CRISPR/Cas9-mediated loss of Rnf150 function did not cause any overt phenotype, nor did it rescue loss of Rnf149-r function, leading us to rule out Rnf150 as a mediator of the *Rnf149-r^CRISPR^* phenotype (Appendix A). 

### 2.3. Rnf149-r Regulates Cardiopharyngeal Fates Independently of FGF/MAPK Signaling

Fibroblast growth factor/mitogen-activated protein kinase (FGF/MAPK) signaling is a key regulator of cardiopharyngeal fates in *Ciona*, with established roles in early *Mesp*+ mesoderm specification and multipotent progenitor induction and migration [12,18]. Sustained FGF/MAPK activity leads to localized *Ebf* expression in ASM precursors, while its exclusion from first and second heart precursors permits cardiac specification [19,22]. MAPK activity in early pharyngeal muscle progenitors initiates *Ebf* expression, until Ebf accumulation permits MAPK-independent auto-activation. This switch is surmised to explain the transition from the multipotent state to the committed pharyngeal muscle fate [19].

The *Rnf149-r^CRISPR^* phenotype resembles the loss of MAPK function, as observed following lineage-specific misexpression of a dominant-negative form of the FGF receptor, or by treatment with the MEK inhibitor U0126 (Razy-Krajka et al., 2018). Moreover, in vitro studies have shown that human RNF149 interacts with and induces ubiquitination of the classic regulator of Mek1/2 and MAPK signaling, *Braf* [34]. We thus hypothesized that *Rnf149-r* regulates the pharyngeal muscle fate choice by interacting with FGF/MAPK signaling. 

In order to test these hypotheses, we overexpressed constitutively active forms of M-Ras and Mek1/2 in parallel with *Rnf149-r*^CRISPR^ and used *Ebf* expression as the readout of pharyngeal muscle fate specification. The overexpression of constitutively active forms of either M-Ras or Mek1/2 suffices to cause ectopic *Ebf* expression in the cardiopharyngeal lineage and abolish the heart fate [19] (Figure 3). We first combined *Rnf149-r*^CRISPR^ with overexpression of a defined constitutively active form of M-Ras, M-Ras^G22V^ (called M-Ras^CA^ hereafter), which is the only Ras homolog in *Ciona* and acts in the FGF/MAPK pathway [35]. We also overexpressed a constitutively active form of Mek1/2, Mek1/2^S220E,S216D^ (Mek^CA^ hereafter) [19], a key regulator of MAPK activity downstream of M-Ras (Figure 3a). We expressed these constructs using the TVC-specific *Foxf* enhancer to restrict the misexpression of the constitutively active mutants to the TVCs and their progeny. Accordingly, we did not observe any unrelated early cardiopharyngeal development defects. In either case, the effects of *Rnf149-r*^CRISPR^ dominated the ectopic activation of the Ras/Mek pathway and blocked *Ebf* expression (Figure 3). The dominance of the *Rnf149^CRISPR^* phenotype was even more clearly observable when using *Foxf>LacZ* to label transfected cells and account for mosaicism (Appendix A). These results indicate that Rnf149-r function is required, either in parallel to the FGF/MAPK pathway or downstream of Mek, for proper *Ebf* expression and by extension for pharyngeal muscle specification. 

The transcription factor *Ets1/2* is a known downstream effector of the FGF/MAPK pathway, presumed to control cardiopharyngeal development in *Ciona* [18,36] (Christiaen lab, unpublished observations). We tested possible functional interactions between *Rnf149-r* and *Ets1/2*, finding that *Rnf149-r*^CRISPR^ also inhibits the ectopic *Ebf* expression phenotype obtained with *Ets1/2* overexpression (Figure 3). 

This systematic dominance of the *Rnf149-r^CRISPR^* phenotype over a gain of either M-Ras, Mek1/2, or Ets1/2 function suggested that the uncharacterized protein Rnf149-r acts in parallel to the FGF/MAPK pathway upstream of *Ebf* activation during pharyngeal muscle specification. This is consistent with the above conclusion that Rnf149-r functions later than the late tailbud stage, since FGF/MAPK is already active and necessary for multipotent progenitor induction and maintenance [18,19].

### 2.4. Rnf149-r Regulates Both MAPK-Dependent and Independent Genes

In order to explore the broader transcriptional impact of *Rnf149-r* loss-of-function, we performed lineage-specific bulk RNA-seq experiments on FACS-purified cardiopharyngeal cells following CRISPR/Cas9-mediated mutagenesis of either *Rnf149-r* or *tyrosinase* as a control, in biological triplicate. Out of 15,232 genes quantified, 190 were significantly differentially expressed, with a false discovery rate (FDR) lower than 0.05. Out of 190, 166 of these genes were upregulated and 24 were downregulated in the *Rnf149-r^CRISPR^* condition compared to *Tyr^CRISPR^* controls. *Rnf149-r*, as well as three known pharyngeal muscle progenitor cell-specific markers, namely *Ebf*, *Htr7*, and *Tbx1/10*, were all significantly downregulated in the *Rnf149-r*^CRISPR^ condition [19,22] (Figure 4a, Appendix A). By contrast, the classic cardiac determinants *Nk4/Nkx2-5*, *Gata4/5/6*, and *Hand* and the heart precursor markers *Slit*, *Lrp4/8*, and *Mmp21* were slightly upregulated but not significantly (Appendix A).

As MAPK is a key regulator of fate in the cardiopharyngeal lineage, we compared the significantly changing expression levels with those observed following overexpression of the dominant-negative Fgf receptor, *Foxf>Fgfr^DN^*. We observed a positive correlation between the fold-changes for both experiments when only the significantly changing genes in the *Rnf149-r*^CRISPR^ experiment were considered (Figure 4b). There was a significant correlation between the transcriptome responses to the loss of Rnf149-r and Fgfr functions. A Fisher’s test comparing the RNA-seq datasets obtained following *Fgfr*^DN^ and *Rnf149-r^CRISPR^* perturbations showed a greater overlap of genes that were significantly dysregulated than expected by chance (*p*-value = 4.7 × 10^−08^, odds ratio = 3.3) (Figure 4c). Since *Fgfr^DN^* has been previously shown to cause the upregulation of cardiac markers [22], these observations are consistent with a partial conversion of pharyngeal muscle progenitors to a heart-like fate in *Rnf149-r*^CRISPR^. 

## 3. Discussion

In this study, we identified ring finger 149 related (*Rnf149-r*) as a new regulator of cardiopharyngeal lineage development in the tunicate *Ciona*. We showed that the predicted Rnf149-r sequence contains a protein–protein interaction domain typically found in other Ring finger ubiquitin ligases, and that CRISPR/Cas9-mediated loss-of-function affects pharyngeal muscle fate specification. We developed molecular tools to study the function of this gene using CRISPR/Cas9 reagents and epistasis assays via overexpression and expression of dominant-negative reagents altering the activity of the FGF/MAPK pathway. Our analyses suggested that *Rnf149-r* acts in parallel to the FGF/MAPK pathway on shared targets. We thus uncovered a potential entry point for a novel pathway regulating cardiac vs. pharyngeal muscle fate specification.

Recent studies from our lab have shown that transcriptional inputs from FGF/MAPK signaling are required at successive stages for pharyngeal muscle specification in *Ciona* [19,22]. CRISPR/Cas9-mediated loss of *Rnf149-r* function phenocopied the loss of *Ebf* expression and pharyngeal muscle specification induced by inhibition of FGF/MAPK signaling, and an RNF149 homolog was shown to regulate Raf; we thus hypothesized that *Rnf149-r* regulates MAPK signaling. However, *Rnf149-r* loss of function did not alter expression of the multipotent progenitor marker *Hand-r*, the maintenance of which relies on continuous inputs from MEK activity. In addition, while lineage-specific bulk RNA-seq analysis of either *Rnf149-r^CRISPR^* or Fgfr inhibition showed a correlated and significant overlap of differentially expressed genes, including known STVC and ASMF markers such as *Htr7*, *Tbx1/10*, and *Ebf*, there were substantial fractions of genes dysregulated by perturbation of either FGF-MAPK or Rnf149-r alone. Moreover, functional interaction assays between *Rnf149-r^CRISPR^* and gain of Ras, Mek, and Ets1/2 functions, indicated that Rnf149-r activity was required for each gain-of-function perturbation to cause ectopic *Ebf* expression, suggesting that *Rnf149-r* acts in parallel with FGF/MAPK/Ets, targeting a partially shared set of genes. 

We note several possible future extensions of this work. First, as *Rnf149-r* is a primed heart gene, it might itself be subject to post-transcriptional regulation. Second, the role of the protein interaction domain in *Rnf149-r* is unknown, and future pulldown experiments followed by mass spectrometry-based identification of interaction partners would provide insights into Rnf149-r molecular function and the hypothesized regulatory pathway involved. 

## 4. Methods

### 4.1. Ciona Robusta Handling

Wild and gravid *Ciona robusta*, also known as *Ciona intestinalis* type A, adults were obtained from M-REP (Carlsbad, CA, USA) and kept under constant light to avoid spawning. Gametes from several animals were collected separately for in vitro cross-fertilization followed by dechorionation and electroporation as previously described [37]. The embryos were cultured in filtered artificial seawater buffered with TAPS (FASW-T) in agarose-coated plastic Petri dishes at 18 °C. We electroporated 50 µg of construct for FACS purification (*Mesp>tagRFP*, *MyoD905>eGFP* and *Hand-r>tagBFP*) and 70 µg of experimental construct (*Mesp>LacZ*, *Mesp>Fgfr^DN^*, *Mesp>Mek^S216D,S220E^*).

### 4.2. CRISPR/Cas9-Mediated Mutagenesis

Six to eight single guide RNAs (sgRNA) per gene with Doench scores (http://crispor.tefor.net, v4.0, accessed on 12 November 2022) [29] higher than 60 were designed to induce mutagenesis using CRISPR/Cas9 in the B7.5 lineage as described [30] (Appendix A). The efficiency of sgRNAs was evaluated using the peakshift method as described [30]. CRISPR/Cas9-mediated deletions were also evaluated by PCR amplification directly from embryo lysates following electroporation with *Ef1a>nls::Cas9-Gem::nls*. sgRNAs were expressed using the *Ciona robusta U6* promoter [28]. For each gene, two or three guide RNAs were used to total 50 μg, in combination with 25 μg of each expression plasmid. A total of 25 μg of Mesp>nls::Cas9-Gem::nls plasmid was co-electroporated with guide RNA expression plasmids for B7.5 lineage-specific CRISPR/Cas9-mediated mutagenesis. One guide RNA was used to mutagenize *tyrosinase* and *neurogenin*, which are not expressed in the cardiopharyngeal lineage and thus were used to control the specificity of the CRISPR/Cas9 system [22].

### 4.3. Molecular Cloning of Rnf149-r^mut^ Rescue Construct

The coding sequence for wild-type *Rnf149-r* (KH.C2.994) was obtained from the plasmid contained in the *C. intestinalis* full ORF Gateway-compatible clone VES66_B12. Insertion of the product into the expression vector was performed using the In-fusion protocol (Clontech, Mountain View, CA). Oligonucleotide-directed mutagenesis and two-step overlap PCRs were used to generate the point-mutated form *Rnf149-r*^mut^ from the corresponding wild-type sequences. We used oligonucleotide-directed mutagenesis to generate mismatches in the PAM sequences adjacent to the sgRNA targets. As a result of the disturbance of a correct PAM sequence (NGG, (reverse complement CCN)), overexpressed *Rnf149-r*^mut^ is resistant to Cas9 nuclease activity.

### 4.4. Fluorescent In Situ Hybridization Immunohistochemistry (FISH-IHC) of Ciona Embryos

The following ISH probes were obtained from plasmids contained in the *C. intestinalis* full ORF Gateway-compatible clone: *Rnf149-r* (VES66_B12), *Bag3/4* (VES90_E04), *Rbm24/38* (VES87_P24), *Asb2* (VES74_P16), *Qki* (VES69_A06), *Rbms1/2/3* (VES90_E05), *Psmd14* (VES70_A18), *Ube2ql1* (VES68_H10), *Otud3* (VES70_G23), *Natn1* (VES91_K15) and: *C. intestinalis* gene collection release I: *Smurf1/2* (GC20g07), *Pcbp3* (GC07e08), *Ube2j1* (GC03l08). 

PCR amplification of transcription templates was performed with the following oligos: M13 fw (5′-GTAAAACGACGGCCAGT-3′) and M13 rev (5′-CAGGAAACAGCTATGAC-3′). DIG- and fluorescein-labeled probes were transcribed with T7 RNA polymerase (Roche) using DIG labeling (Roche) and purified with the RNeasy Mini Kit (Qiagen). Antisense RNA probes were synthesized as described [38]. In vitro antisense RNA synthesis was performed using T7 RNA polymerase (Roche, Cat. No. 10881767001) and the DIG RNA Labeling Mix (Roche, Cat. No. 11277073910). 

Embryos were harvested and fixed at desired developmental stages for 2 h in 4% MEM-PFA (4% paraformaldehyde, 0.1 M MOPS pH 7.4, 0.5 M NaCl, 1 mM EGTA, 2 mM MgSO4, 0.05% Tween 20), rinsed in cold phosphate-buffered saline (PBS), gradually dehydrated for 1.5 h, and stored in 75% ethanol at −20 °C. They were then rehydrated gradually using a methanol/PBS–Tween series, and whole-mount fluorescence in situ hybridization was performed as previously described [16,17]. An anti-digoxigenin-POD Fab fragment (Roche, Indianapolis, IN, USA) was first used to detect the hybridized probes, then the signal was revealed using tyramide signal amplification (TSA) with the Fluorescein TSA Plus Evaluation Kit (Perkin Elmer, Waltham, MA, USA). 

For immunohistochemistry, samples were blocked in Tris–NaCl blocking buffer (Blocking Reagent, PerkinElmer) for 2–4 h preceding primary antibody incubation and 1 h preceding secondary antibody incubation. Antibody solutions were prepared in Tris–NaCl blocking buffer and incubated for 1–2 h at room temperature, followed by an overnight incubation at 4 °C. Anti–β-galactosidase monoclonal mouse antibody (Promega, 1:500) was co-incubated with anti-mCherry polyclonal rabbit antibody (Bio Vision, Cat. No. 5993–100, 1:500) for immunodetection of *Mesp>nls::lacZ* and *Mesp>hCD4::mCherry* products, respectively. Goat anti-mouse secondary antibodies coupled with AlexaFluor-555 and AlexaFluor-633 were used to detect β-galactosidase-bound mouse antibodies and mCherry-bound rabbit antibodies after the TSA reaction. Antibody washes were performed using Tris–NaCl–Tween buffer. Samples were mounted in ProLong Gold Antifade Mountant (Thermo Fisher Scientific, Waltham, MA, USA, catalog number P36930) and stored at 4 °C.

Images were acquired with an inverted Leica TCS SP8 X confocal microscope, using an HC PL APO ×63/1.30 objective. Maximum projections were processed with maximum projection tools from the Leica software LAS-AF.

### 4.5. Cell Dissociation and FACS-Purification of Ciona Robusta Cells

Sample dissociation and FACS were performed as previously described [39,40]. Embryos and larvae were harvested at 15 hpf in 5-mL borosilicate glass tubes (Fisher Scientific, Waltham, MA. Cat. No. 14-961-26) and washed with 2 mL calcium- and magnesium-free artificial seawater (CMF-ASW: 449 mM NaCl, 33 mM Na_2_SO_4_, 9 mM KCl, 2.15 mM NaHCO_3_, 10 mM Tris-Cl pH 8.2, 2.5 mM EGTA). Embryos and larvae were dissociated in 2 mL 0.2% trypsin (*w*/*v*, Sigma, St. Louis, MO, USA, T- 4799) in CMF-ASW by pipetting with glass Pasteur pipettes. The dissociation was stopped by adding 2 mL filtered ice cold 0.05% BSA CMF-ASW. Dissociated cells were passed through a 40-μm cell-strainer and collected in a 5-mL polystyrene round-bottom tube (Corning Life Sciences, Oneonta, New York, NY, USA). Cells were collected by centrifugation at ~5.34 × 10^−10^ m^3^⋅kg^−1^⋅s^−2^ (i.e. 800 G) for 3 min at 4 °C, followed by two washes with ice cold CMF-ASW. Cell suspensions were filtered again through a 40 μm cell-strainer and kept on ice. 

Cardiopharyngeal lineage cells were labeled with *Mesp>tagRFP* and *Hand-r>tagBFP* reporters. The mesenchyme cells were counter-selected using *MyoD905>GFP*. Dissociated cells were loaded in a BD FACS AriaTM cell sorter. A 488-nm laser and FITC filter were used for GFP; a 407-nm laser and DsRed filter were used for tagRFP; and a 561-nm laser and Pacific BlueTM filter were used for tagBFP.

### 4.6. RNA-seq Library Preparation, Sequencing and Analysis

To profile the transcriptomes of FACS-purified cells from *Rnf149-r^CRISPR^* and control samples, 1,000 cells were directly sorted in 100 μL lysis buffer from the RNAqueous-Micro Total RNA Isolation Kit (Ambion, Waltham, MA, USA). For each condition, samples were obtained in three biological replicates. The total RNA extraction was performed following the manufacturer’s instruction. The quality and quantity of total RNA was checked using Agilent RNA ScreenTape (Agilent, Santa Clara, CA, USA) using the 4200 TapeStation system. RNA samples with an RNA integrity number (RIN) >8 were kept for downstream cDNA synthesis. A total of 250–2000 pg of total RNA was loaded as a template for cDNA synthesis using the SMART-Seq v4 Ultra Low Input RNA Kit (Clontech, Mountain View, CA, USA) with template switching technology. RNA-Seq libraries were prepared and barcoded using the Ovation Ultralow System V2 (NuGen, San Carlos, CA, USA). Six barcoded samples were pooled in one lane of the flow cell and sequenced by Illumina NextSeq 750 (MidOutput run). Paired-end 75-bp-length reads were obtained from all the bulk RNA-seq libraries. Bulk RNA-seq libraries were aligned using STAR 2.7.0a [41] with the parameters ‘--runThreadN 6--outSAMtype BAM SortedByCoordinate\--outSAMunmapped Within\--outSAMattributes Standard’. Counts were obtained using featureCounts, a function of subread [42,43]. Differential expression was calculated using DESeq2 [44].

### 4.7. Data Availability

The RNA-seq data were deposited in the Gene Expression Omnibus (GEO) under accession GSE171152. Previously published bulk RNA-seq data that were used here for comparison in Figure 4 can be found on GEO under accession GSE99846.

## Figures and Tables

**Figure 1 ijms-24-08865-f001:**
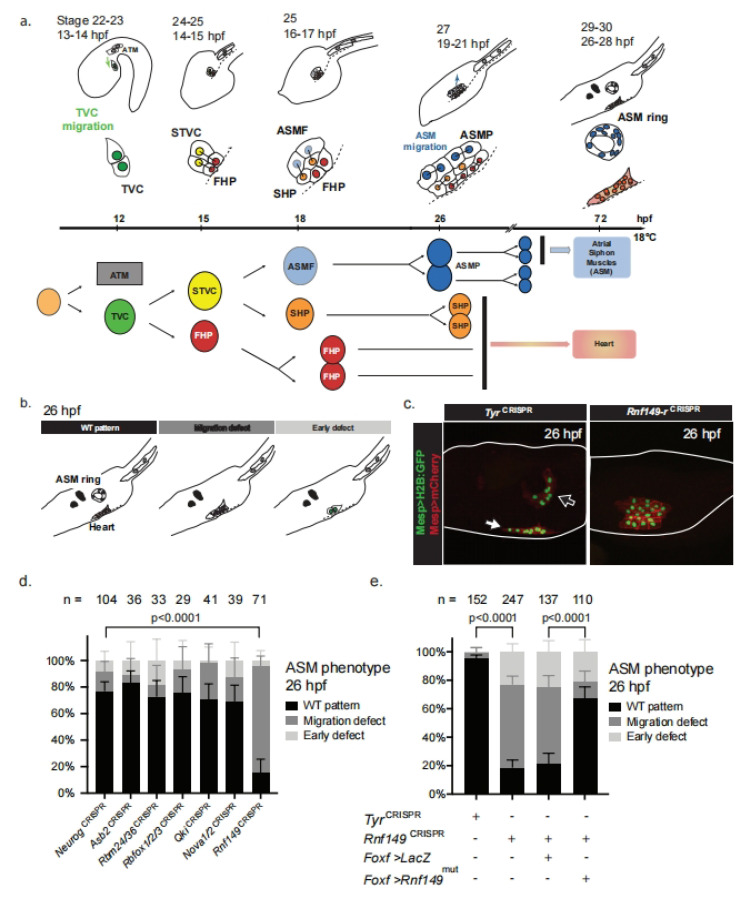
*Rnf149-r*^CRISPR^ causes ASM migration defects. (**a**) Schematic of *Ciona* development showing asymmetric cell divisions and resulting cell fates of the cardiopharyngeal mesoderm (CPM). Stages are set according to Hotta et al. (2007) with hours post fertilization (hpf) at 18 °C (Appendix A) [20,21]. Anterior tail muscle (ATM, gray), trunk ventral cell (TVC, green), secondary TVC (STVC, yellow), first heart precursor (FHP, red), second heart precursor (SHP, orange), atrial siphon muscle founder cells (ASMF, blue). Dashed lines indicate the ventral midline. (**b**) Schematic representation of phenotypes scored in Figure 1c. (**c**) Tyr^CRISPR^ used as control. Cardiopharyngeal lineage cells are marked by mCherry and GFP driven by *Mesp*. H2B::GFP (green) and hCD4::mCherry (red) accumulate in the nuclei and at the cell membrane, respectively. Arrow indicates the heart progenitors and the open arrow indicates the ASM ring. (**d**) Histogram with phenotype proportions. *Neuro*g^CRISPR^ are used as control, a gene that is known to be inactive in the cardiopharyngeal lineage. First 5 genes scored compared to *Neurog*^CRISPR^ did not show significant differences (Fisher exact test), while *Rnf149-r*^CRISPR^ differed from controls with *p*-value < 0.0001. Experiments were performed in biological replicates and “n=” represents the total numbers of individual halves scored per condition. Error bars represent 95% Wilson method of confidence interval for proportions. (**e**) Corresponding histogram with phenotype proportions. Experiments are performed in biological replicates. “n=” represents the total numbers of individual halves scored per condition. Error bars represent 95% Wilson method of confidence interval for proportions.

**Figure 2 ijms-24-08865-f002:**
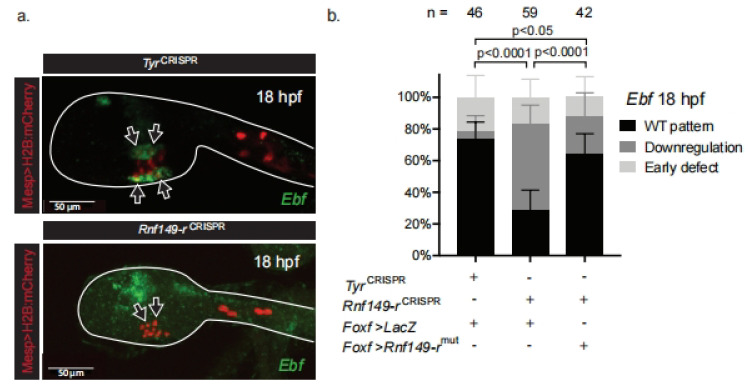
*Rnf149-r*^CRISPR^ causes morphogenetic defects in the cardiopharyngeal lineage. (**a**) In situ hybridization using an *Ebf* probe (green). *Tyr*^CRISPR^ used as control. Cardiopharyngeal lineage are marked by mCherry driven by *Mesp* and revealed by mCherry antibody in red. H2B::mCherry accumulates in the nuclei. Scale bar is 50 μm. Arrows mark ASMFs. (**b**) Corresponding histogram with phenotype proportions. “n=” represents the total numbers of individual halves scored per condition. Error bars represent 95% Wilson method of confidence interval for proportions.

**Figure 3 ijms-24-08865-f003:**
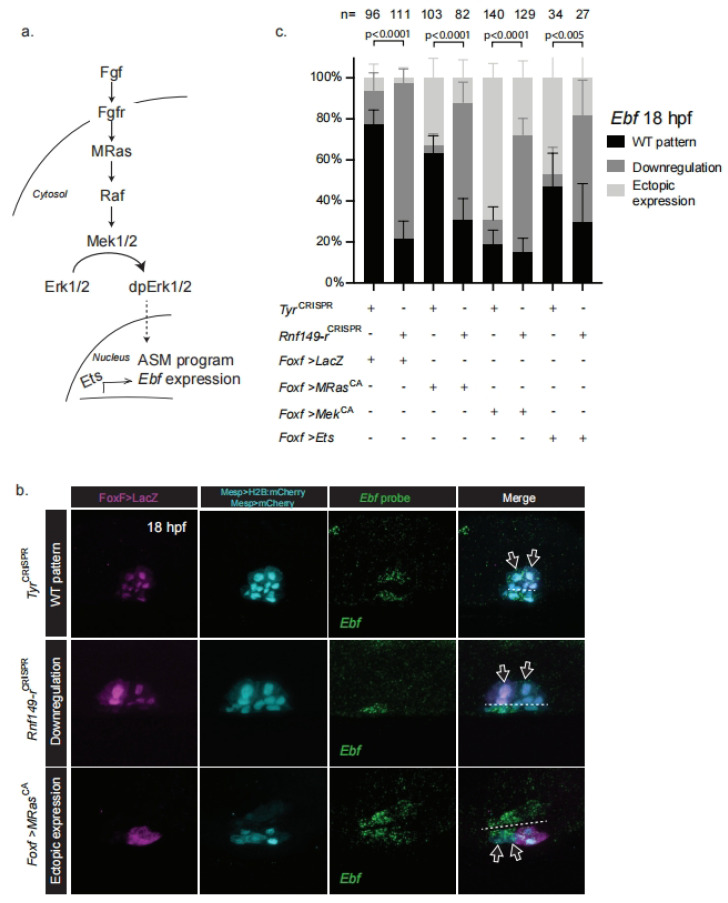
*Rnf149-r* acts in parallel to the FGF/MAPK pathway. (**a**) Schematic representation of the FGF/MAPK pathway. (**b**) In situ hybridization using an *Ebf* probe. *Tyr*^CRISPR^ are used as control. Cardiopharyngeal lineage cells are marked by mCherry and beta-galactosidase/LacZ driven by *Mesp* and *FoxF*, revealed by anti mCherry and anti beta-galactosidase antibodies in magenta and cyan, respectively. H2B::mCherry and hCD4::mCherry (cyan) accumulate in the nuclei and at the cell membrane, respectively. Dashed lines indicate the ventral midline. Arrows mark the Atrial Siphon Muscle Founder cells (ASMFs). (**c**) Corresponding histogram with phenotype proportions. “n=” represents the total numbers of individual halves scored per condition. Error bars represent 95% Wilson method of confidence interval for proportions.

**Figure 4 ijms-24-08865-f004:**
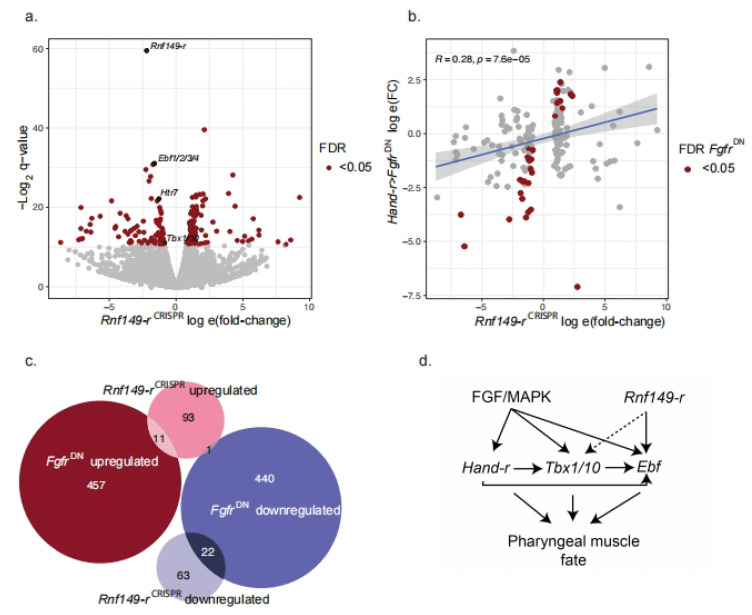
*Rnf149-r* has novel targets contributing to the cell fate choices in the cardiopharyngeal lineage. (**a**) Volcano plot showing significantly upregulated and downregulated genes upon *Rnf149-r*^CRISPR^ compared to control (*Tyr*^CRISPR^) cardiopharyngeal lineage cells in red (FDR < 0.05). Genes shown in black are well-studied ASM-specific markers (*Ebf1/2/3/4*, *Htr7* and *Tbx1/10*), and our target gene, *Rnf149-r*. Experiments are done in triplicates. (**b**) Log fold-change correlations between MAPK inhibition (*Hand-r>Fgfr^DN^*) (Wang et al., 2019) and *Rnf149-r*^CRISPR^, using only differentially expressed genes in either condition. Genes shown in red are differentially expressed in the *Rnf149-r*^CRISPR^ condition. R represents the Pearson correlation coefficient. (**c**) Euler diagram showing overlaps of differentially expressed gene groups in *Rnf149-r*^CRISPR^ and *Hand-r>Fgfr^DN^*. Mutual enrichment *p*-value, calculated using Fisher’s exact test, is 4.7 × 10^−8^. Odds ratio is calculated to be 3.3. (**d**) Proposed model of *Rnf149-r*’s effect on regulation of the cardiopharyngeal lineage in *Ciona*.

## Data Availability

The RNA-seq data were deposited in the Gene Expression Omnibus (GEO) under accession GSE171152. Previously published bulk RNA-seq data that were used here for comparison in Figure 4 can be found on GEO under accession GSE99846.

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
