# Peer review of "Ring Finger 149-Related Is an FGF/MAPK-Independent Regulator of Pharyngeal Muscle Fate Specification"

_ijms, 2023, doi:10.3390/ijms24108865_

Round 1
Reviewer 1 Report
The authors pose a reasonable biological question concerning the differentiation of cardiopharyngeal mesoderm, and test a limited number of candidate genes encoding post translational modyfiers using a standard and well-designed crispr-cas9 strategy. They land on a gene, Rnf149-r, the mutation leads to down regulation of the critical Ebf and Tbx1/10 genes and perturbation of the pharyngeal muscle specification/development. This effect appears to be substantially independent from the FGF/MAPK, opening an interesting window over alternative modes of regulation of Ebf.
A number of question will need to be addressed in future experiments, for example: is Ebf expression sufficient to rescue the Rnf149r-dependant phenotype? Is the Rnf149-r mutation effect a transcriptional or post-transcriptional phenomenon? Which are the targets of the hypothesized ubiquitin-ligase function of the Rnf149r-encoded protein?
Nevertheless, the work is meritorious, rigorous, and of interest for the specific field and for the developmental biology community.
Comments
1) The sentence (ln 296) is confusing. "The upregulated genes in the RNA-seq were significantly enriched in FGF-MAPK inhibited genes and the downregulated genes are significantly enriched in FGF-MAPK-dependent genes". Could the authors specify the difference between "downregulated genes" and "FGF-MAPK-dependent genes"? (also, correct the tense of the sentence).
2) The conclusion (ln. 303): " These observations are consistent with a partial conversion of pharyngeal muscle progenitors to a heart-like fate in Rnf149-rCRISPR" seems to come out of the blue and is unsubstantiated because the relevant markers do not change significantly. Therefore, it should be removed. Indeed, within the sentence (ln. 282) "...were slightly upregulated..." should also be removed because the change is not significant.
Author Response
Thank you for the careful consideration and thoughtful suggestions. We will correct the manuscript accordingly, but will not be able to perform additional experiments, especially the aspects that the reviewer rightfully identified as future work.
Best regards
Lionel Christiaen

Reviewer 2 Report
Cardiopharygeal mesoderm includes multipotent progenitor cells, which give rise to branchiomeric muscles in the head and cardiomyocytes in the heart in mammals. Intriguingly, a similar cell population called cardiopharygeal field is found in the invertebrate chordate Ciona robusta, and due to the simple anatomy and development of the animal, understanding of the field is growing at the cellular and molecular levels. Vitrinel et al., performed detailed and rigorous examinations of Ciona cardiopharygeal field and provided interesting data on the specification of the field. I have only few comments on the manuscript.
Page 1, line 21
Not sure this is a correct expression. “CRISPR/Cas9-mediated loss of Rnf149-r function” would be suitable.
Page 4, line 101
It is better to describe the formal name of Rnf149-r, Ring finger protein 149 related, in the Introduction.
Page 5, line 144
Figure 1a, line 67
“cardiopharyngeal mesoderm” would be cardiopharyngeal progenitors, lineages, or field.
Figure 1b legend
Please describe the details of migration and early defects.
Figure 2a
Ebf expression is observed in the head and neck part of Ciona larva after Rnf149-r CRISPR/Cas9mutagenesis, which is not found in the control larva. Please provide an explanation for the expression.
Author Response
Thank you for the careful consideration and thoughtful suggestions. We will correct the manuscript accordingly.
Best regards
Lionel Christiaen

Reviewer 3 Report
In their manuscript, Vitrinel et al. seek to understand cardiac vs pharyngeal muscle fate decision. They have looked for post-transcriptional mechanisms and have selected 14 candidate genes (based on their expression and the activity of their proteins). They have conducted lineage-specific CRISPR/Cas9 for 5 of these genes. They have fund that loss of Rnf149-r disrupted pharyngeal muscle development by inhibiting the expression of Ebf, a key player for pharyngeal muscle cell fate decision. The phenotype, described by the authors, phenocopies the loss of MAPK signaling pathway. Furthermore, the authors have demonstrated that Rnf149-r regulates some MAPK dependent genes but acts in parallel of the signaling pathway. Altogether, the study by Vitrinel and colleagues has identified Rnf149-r, as a new regulator of pharyngeal muscle fate decision.
Overall, this manuscript is important for the field of cardiopharyngeal mesoderm development, with the identification of a new player for pharyngeal muscle fate decision. In general, the manuscript is very well written and clear. We strongly encourage its publication.
Please find below minor suggestions for improvement of the manuscript:
1- The authors have showed that the loss of Rnf149-r led to a downregulation of Ebf and subsequently to the disruption of pharyngeal muscles. This could be rescued with a CRISPR/Cas9-resistant form of Rnf149-r cDNA. Does the overexpression of this CRISPR/Cas9-resistant form of Rnf149-r cDNA has an effect on the cardiac or ASM phenotype?
2- The controls used for the CRISPR/Cas9 experiments are not all consistent and clear. For example, in lines 143-145, NeurogeninCRISPR is used as a control while the figure 1c shows use of TyrCRISPR. This could be easily by writting from the beginning that two types of controls are used.
3- The legends of Fig S2 and S3 should be inverted.
4- It would be useful to add the names of the proteins in Fig S5 and to highlight Rnf149 in such context.
Author Response

(The authors gave the same response as above.)
